# The Advertising Policies of Major Social Media Platforms Overlook the Imperative to Restrict the Exposure of Children and Adolescents to the Promotion of Unhealthy Foods and Beverages

**DOI:** 10.3390/ijerph17114172

**Published:** 2020-06-11

**Authors:** Gary Sacks, Evelyn Suk Yi Looi

**Affiliations:** Global Obesity Centre (GLOBE), Institute for Health Transformation, Deakin University, Burwood, Victoria 3125, Australia; evelyn.looi@deakin.edu.au

**Keywords:** social networks, advertising policy, marketing, digital platforms, unhealthy food

## Abstract

There have been global calls to action to protect children (aged <18) from exposure to the marketing of unhealthy foods and beverages (‘unhealthy foods’). In this context, the rising popularity of social media, particularly amongst adolescents, represents an important focus area. This study aimed to examine the advertising policies of major global social media platforms related to the advertising of unhealthy foods, and to identify opportunities for social media platforms to take action. We conducted a desk-based review of the advertising policies of the 16 largest social media platforms globally. We examined their publicly available advertising policies related to food and obesity, as well as in relation to other areas impacting public health. The advertising policies for 12 of the selected social media platforms were located. None of these platforms adopted comprehensive restrictions on the advertising of unhealthy foods, with only two platforms having relevant (but very limited) policies in the area. In comparison, 11 of the 12 social media platforms had policies restricting the advertising of alcohol, tobacco, gambling, and/or weight loss. There is, therefore, an opportunity for major social media platforms to voluntarily restrict the exposure of children to the marketing of unhealthy foods, which can contribute to efforts to improve populations’ diets.

## 1. Introduction

In 2020, a major report on child health and wellbeing, published by a joint World Health Organization (WHO)–United Nations Children’s Fund (UNICEF)–Lancet Commission, called for strong regulatory action to protect children (aged 0–18 years) from the marketing of unhealthy products, such as tobacco, alcohol and sugar-sweetened beverages [1]. This follows persistent global recommendations from the WHO for member states, the private sector and civil society organisations to restrict the marketing of food and non-alcoholic beverage products high in saturated fats, trans fatty acids, free sugars and/or salt (hereafter referred to as ‘unhealthy foods’) to children and adolescents globally [2,3]. While some governments and food manufacturers have adopted measures to reduce the exposure of children to the marketing of unhealthy foods, these actions are typically not comprehensive in nature, and, globally, much more needs to be done to protect children from the harmful effects of unhealthy food marketing [4,5].

Companies in the food industry, including food and beverage manufacturers, quick-service restaurants and supermarket retailers, use a wide variety of channels and techniques to promote their brands and products as part of integrated marketing communications (IMC) [6,7]. In recent years, social media marketing has grown rapidly to be one of the preferred marketing channels across a range of industries, including the food industry [7,8,9]. Indeed, social media marketing was estimated to account for 13% of the global advertising spend in 2019 [9]. The increased use of social media marketing has facilitated highly tailored marketing communications to reach a wider audience, including children and adolescents [8]. As the number of social media users exceeded 3.8 billion in January 2020 [10], social media advertising spending is set to continue to rise beyond the estimated USD 84 billion spent in 2019 [9].

China is reported to be the world’s largest social network market [11], with 673.5 million social network users in 2018 [11], including approximately 70% who were adolescents aged between 15 and 19 [12]. A recent study conducted in Australia found that 72.9% of the children and adolescents surveyed used the internet for social networking, including 88.6% of adolescents aged between 13 and 16 [13]. The study also reported that more than half (52.6%) of those children who were active on Facebook ‘liked’ a fast food brand [13]. In the United States (U.S.), a recent study found that 70% of adolescents (aged between 13 and 17) engaged with food and beverage brands on social media [14]. Fifty-four percent of the adolescents surveyed reported engaging with fast food brands, whilst 50% engaged with brands of sugary drinks, followed by candy (46%) and snacks (45%) [14].

Most major social media platforms require users to be above 13 years old. However, there is a growing number of children under 13 years old owning an account and using social media networks, with or without parental consent [15]. In 2019, The Office of Communications (Ofcom [16]), the United Kingdom’s (UK) communications regulator, reported that 71% of the 12- to 15-year-old children surveyed had a social media profile [16]. The Ofcom report also found that 21% of children between 8 and 11 years old owned a social media profile, and that 1% of children as young as three years old had a social media profile in the UK [16]. In Australia, 46.5% of children aged between 10 and 12 have been found to be active on social media [13]. The high levels of usage of social media among young children, coupled with the increasingly high advertising spend on social media platforms, is potentially concerning from a public health perspective, given the well-documented vulnerability of children and adolescents to the impacts of marketing [8,17].

Globally, most government regulations aimed at reducing the exposure of children to the advertising of unhealthy foods are voluntary in nature, focus on children under 12 years old, and apply primarily to broadcast media (such as television) [4]. However, several jurisdictions have enacted mandatory regulations that restrict the advertising of unhealthy foods to children in non-broadcast media, including on social media [4,18,19,20]. Chile has perhaps the most comprehensive national level restrictions on the marketing of unhealthy foods, but the regulations (implemented in 2016) only apply to children under the age of 14, and the way in which the regulations have been implemented with regards to social media is not clear [21]. In the UK, whilst current regulations on the marketing of unhealthy foods to children apply to social media, amongst several other marketing channels, they only apply to children under the age of 12, and the restrictions on non-broadcast media are less stringent than on broadcast media [22].

The food industry has taken some limited action to restrict the marketing of unhealthy foods to children [6]. Globally, major food industry associations and transnational companies have pledged to reduce advertising to children [4]. However, existing voluntary industry self-regulation has been shown to be ineffective in protecting children from exposure to unhealthy food marketing [1]. Furthermore, given the large number of adolescent social media users, it is concerning that the ages between 13 and 18 years are typically excluded from the existing regulations designed to reduce food marketing to children.

Social media platforms typically generate a large proportion of their revenue from advertising [9]. Advertising on social media is primarily self-regulatory, whereby businesses that market themselves on social media platforms are responsible for the accuracy of contents and for complying with local laws and regulations [23]. In addition, social media platforms typically apply their own advertising policies which govern the types of advertising that are permitted on their platform. Major social media platforms, such as Facebook and YouTube, have shown a willingness to restrict the advertising of products that may be harmful to public health, such as in the areas of tobacco, alcohol and gambling [24,25]. However, the policies of major social media platforms regarding the advertising of unhealthy foods has not been previously examined, and their potential role in restricting the exposure of children to unhealthy foods has largely escaped attention.

Given the global imperative to protect children from exposure to unhealthy food marketing, and the weaknesses in the current regulation of unhealthy food marketing on social media from government and food industry stakeholders, this study aimed to examine the advertising policies of major global social media platforms related to the advertising of unhealthy foods, and to identify opportunities for social media platforms to take action to limit the exposure of children to the marketing of unhealthy foods. We hypothesised that social media platforms were not taking sufficient action to restrict the promotion of unhealthy foods.

## 2. Materials and Methods

### 2.1. Sample Selection

This study utilised an exploratory approach to identify existing policies of major social media platforms (hereafter known as ‘platforms’), related to the advertising of unhealthy products and/or services. The 16 most popular platforms (based on the number of active users as of October 2019, as per data from Statista [26]), were selected for analysis. These were, in order of popularity: (1) Facebook; (2) YouTube; (3) WhatsApp; (4) Facebook Messenger; (5) WeChat; (6) Instagram; (7) QQ; (8) QZone; (9) Douyin/Tik Tok; (10) Sina Weibo; (11) Reddit; (12) Twitter; (13) Douban; (14) Snapchat; (15) LinkedIn; and (16) Pinterest.

### 2.2. Data Collection

A desk-based review of each platform’s primary website (obtained from an internet search) was conducted to identify the existing advertising policy provisions adopted by the selected platforms relating to food and obesity. Each platform’s advertising policy provisions concerning four other areas of public health relevance (topic areas), namely tobacco, alcohol, gambling and weight loss, were examined as comparators. These topic areas were selected after an initial review of available advertising policy provisions revealed that many of the platforms had relevant policy provisions in these areas.

For each included policy and topic area, the following information was extracted: (i) name of the platform; (ii) classification of the platform; (iii) minimum age to create an account; (iv) advertising provisions relating to the topic area with respect to all age groups; and (v) advertising provisions relating to the topic area with respect to children specifically. In addition, where advertising policies contained provisions with explicit age-related conditions, these were extracted and categorised separately. All policy information was extracted in December 2019. For all but one platform (WeChat), policies were available in English. With respect to WeChat, their policy was available in Mandarin, and was translated from Mandarin into English by one of the authors (E.S.Y.L.) for analysis.

### 2.3. Data Analysis

The data were analysed descriptively. In analysing the data, the term ‘minors’ was used to refer to any person below the age of eighteen, unless specified otherwise. Policies in which the platform prohibited advertising (ads) in a certain area were classified as ‘prohibited’, whilst policies that required prospective advertisers to seek prior approval from the platform, or to make a commitment to comply with relevant rules and regulations of governments and/or regulatory entities, were classified as ‘restricted’.

## 3. Results

The characteristics of the 16 selected platforms and their relevant advertising policies are summarised in Table 1, with additional details provided as Appendix A. Advertising policies of four of the 16 selected platforms (i.e., QQ, QZone, Sina Weibo, and Douban) were unavailable.

Ten of the 12 platforms where advertising policies were available had no relevant policies related to unhealthy foods and beverages (see Table 1). Snapchat had a policy requiring advertising pertaining to any food products to provide an accurate description of the characteristics and qualities of food production (e.g., related to nutrition claims made as part of the ad), with no additional restrictions on the advertising of unhealthy foods and beverages. YouTube did not have any restrictions on the advertising of unhealthy foods and beverages on its primary platform. However, on its dedicated children’s platform, YouTube Kids, developed for children under 13 years old, the advertising of products related to consumable foods and beverages was prohibited, regardless of the nutrient content of the products. YouTube Kids also prohibited ads that would incite children to purchase a product/services or urge their parents to do so.

Eleven of the 12 platforms implemented a complete ban on the advertising of tobacco products and related paraphernalia, including chewing tobacco and electronic cigarettes (see Table 1). Douyin/Tik Tok did not have restrictions on ads relating to tobacco products or services. However, Douyin/Tik Tok had a term to protect minors which stipulated that ads could not ‘[harm] minors in any manner’, although there were no further details on how ‘harm’ should be interpreted.

Eight of the 12 platforms restricted the advertising of alcohol and alcohol-related products on their services, whereby the platforms specifically highlighted that advertisements containing alcohol must comply with the laws and regulations of countries and territories, including age-targeting criteria (see Table 1). These platforms also made it explicit that ads related to alcohol must not target minors as defined by local laws and regulations.

Three platforms (WhatsApp, WeChat and LinkedIn) had policies that prohibited gambling ads on their platforms. Nine other platforms specified that ads promoting gambling must comply with local laws and regulations within the countries that the ads are targeting. In addition, five platforms (Facebook, Facebook Messenger, Instagram, Reddit, and Snapchat) required prospective advertisers to obtain the platform’s permission prior to gambling ads or campaigns being aired on their platforms. Five platforms (Facebook, Facebook Messenger, Instagram, YouTube and Twitter) had specific terms prohibiting the advertising of gambling and gambling-related products, and/or services to people aged below 18. YouTube Kids prohibited all ads that were considered both restricted and prohibited contents on its main platform (i.e., YouTube), which included gambling ads.

Five platforms (Facebook, Facebook Messenger, WhatsApp, LinkedIn, and Instagram) prohibited the advertising of weight loss products and/or services, especially ads that generate negative self-perception and/or mislead users into believing the health benefits of such products. Both Facebook and Instagram explicitly stipulated that the advertising of weight loss products must not target any persons below the age of 18. YouTube Kids included a ban on the advertising of weight loss products and/or services.

## 4. Discussion

Despite strong international recommendations for action to reduce the exposure of children to the marketing of unhealthy foods, none of the most widely used social media platforms adopt comprehensive restrictions on the advertising of unhealthy foods. Even though one of the platforms developed for children, YouTube Kids, has an advertising policy prohibiting the marketing of foods on its platform, there is evidence that users of the platform may nevertheless be exposed to unhealthy food brands through product placement and promotional videos on the platform [27]. When coupled with the absence of comprehensive government regulation of the marketing of unhealthy foods, social media platforms represent a high-risk channel for children and adolescents to be exposed to the marketing of unhealthy foods.

In contrast, the vast majority of the most popular social media platforms have policies that prohibit or restrict the advertising of products and/or services relating to alcohol, tobacco, gambling, and/or weight loss. Importantly, in each of these areas, social media platforms’ advertising policies are able to cater to country-level variations in government regulations. For example, alcohol ads cannot target people below the age of 25 in Sweden, whilst the minimum age requirement for alcohol ads in Canada is 19. Additionally, ads promoting and referencing alcohol have been outlawed in Norway, many countries in the Middle East (e.g., Saudi Arabia and United Arab Emirates) and countries with Islam as the state religion (e.g., Brunei and Bangladesh). With respect to weight loss, the policies of multiple social media platforms are more restrictive than most government policies. For example, Instagram and Facebook recently implemented a ban on the advertising of diet and weight loss products, as well as cosmetic procedures, to users below 18 years old [28]. The existence of the observed advertising policies demonstrates both the willingness of, and feasibility for, social media platforms to take action related to health and the protection of children from the advertising of unhealthy products. Future research should examine the drivers behind these policy decisions on different platforms, as well as their financial impact on each platform. Such research could help identify potential leverage points and barriers to change.

Exposure to the marketing of unhealthy products, such as alcohol and tobacco, on social media has been found to be associated with a higher risk of related unhealthy behaviours [29]. With respect to food, adolescent social media users’ responses to unhealthy food advertising posts have been found to be significantly greater than their responses to healthy food posts, indicating the greater impact of unhealthy food ads [30]. Furthermore, exposure to the marketing of unhealthy foods on social media platforms has been shown to increase children’s immediate consumption of the promoted product [31]. This is of concern given that, among adolescents, engagement with food and beverage brands on social media is widespread [14]. Accordingly, there is an urgent need for greater action to restrict the marketing of unhealthy foods on social media.

This study highlights the substantial opportunity for major social media platforms to take action to reduce the exposure of children to the marketing of unhealthy foods. In line with global public health recommendations in this area, social media platforms should be encouraged to adopt advertising restrictions that: (i) apply to all children and adolescents under 18; (ii) cover a wide range of marketing techniques (e.g., advertising, child-directed content, product placement); and (iii) use a comprehensive definition of unhealthy foods and beverages, based on government-endorsed criteria [2,4,8,17,32].

Policies from other global digital platforms provide examples of good practice in this area. For example, two multinational children’s entertainment networks (The Walt Disney Company and Nickelodeon) provide advertising guidelines for the marketing of food products on their platforms (refer to Appendix A, for further details). Nickelodeon requires that ads on its network must not: (i) promote an unhealthy lifestyle, and (ii) encourage the excessive consumption of unhealthy foods [33]. The Walt Disney Company specifies that the advertising of all food products on their network must comply with the company’s nutrition guidelines [34]. The motivations behind the implementation of these policies, mechanisms for effective monitoring and enforcement, and their impact on the networks, advertisers and users warrants further investigation, and may help to inform action from social media platforms.

This study offered the first comprehensive assessment of the advertising policies of the largest social media platforms, with respect to unhealthy foods. The study was a desktop review that relied only on publicly available information regarding the advertising policies of each social media platform. Future research could include direct correspondence with each social media platform to understand if additional relevant policies exist beyond those that are publicly available, and their appetite for change. Due to the nature of the study, we were not able to assess the extent to which existing policies were implemented, and the extent and prevalence of unhealthy food marketing on each social media platform. Future research should explore methods to assess food marketing on different social media platforms and in different countries.

## 5. Conclusions

This study found that major social media platforms do not have comprehensive policies in place to restrict the marketing of unhealthy foods on their platforms. Globally, existing regulations are proving ineffective in protecting children and adolescents from exposure to the digital marketing of unhealthy foods and beverages. There is an opportunity for major social media platforms to voluntarily restrict the exposure of children to the marketing of unhealthy foods, which can contribute to existing efforts to address obesity and improve population diets. Civil society groups, including consumer advocates, the public health community and investors, need to explore ways to highlight this opportunity to social media platforms, and pressure them to take action in this critical area of population health.

## Figures and Tables

**Table 1 ijerph-17-04172-t001:** Advertising policies of sixteen major social media platforms in relation to unhealthy foods, tobacco, alcohol, gambling and weight-loss, as of December 2019.

Platform (Country of Headquarters) ^1^	Classification	Minimum Age to Create an Account	Unhealthy Foods		Tobacco		Alcohol		Gambling		Weight Loss		Other Policies Related to Content Targeted at Minors
			General	Minors ^2^	General	Minors ^2^	General	Minors ^2^	General	Minors ^2^	General	Minors ^2^	
Facebook (United States)	Social Media	13	No restrictions	No restrictions	Prohibited		Restricted—country laws and age requirements apply	Restricted—specific policy to protect under 18s	Restricted	Restricted—specific policy to protect under 18s	Prohibited	Restricted—specific policy to protect under 18s	Not applicable
YouTube (United States)	Video Sharing	18 13 (with parental permission)	No restrictions	No restrictions	Prohibited		Restricted—country laws and age requirements apply	Restricted—specific policy to protect under 18s	Restricted	Restricted—specific policy to protect under 18s	Restricted		Not applicable
		Under 13 (dedicated platform for children, *YouTube Kids*)		Prohibited—Specific policy to protect minors	Prohibited—Specific policy to protect minors		Prohibited—Specific policy to protect minors		Prohibited—Specific policy to protect minors		Prohibited—Specific policy to protect minors		Prohibited—Specific policy to protect minors
WhatsApp (United States)	Messaging	16—from May 2018 13—Prior to May 2018	No restrictions	No restrictions	Prohibited		Prohibited		Prohibited		Prohibited		Not applicable
Facebook Messenger (United States)	Messaging	13	No restrictions	No restrictions	Prohibited		Restricted—country laws and age requirements apply	Restricted—specific policy to protect under 18s	Restricted	Restricted—specific policy to protect under 18s	Prohibited	Restricted—specific policy to protect under 18s	Not applicable
WeChat (China)	Multi-purpose messaging, social media and mobile payment platform	13	No restrictions	No restrictions	Prohibited		No restrictions	No restrictions	Prohibited		No restrictions	No restrictions	Not applicable
Instagram (United States)	Social Media	13	No restrictions	No restrictions	Prohibited		Restricted—country laws and age requirements apply	Restricted—specific policy to protect under 18s	Restricted	Restricted—specific policy to protect under 18s	Prohibited	Restricted—specific policy to protect under 18s	Not applicable
QQ (China)	Social Media	13 Under 12—children’s version available	Not available										
Qzone (China)	Social Media	Not available	Not available										
Douyin/Tik Tok (China)	Social Media	18	No restrictions	No restrictions	No restrictions	No restrictions	No restrictions	No restrictions	Restricted content	Restricted content	No restrictions	No restrictions	Restricted—specific policy to protect minors
Sina Weibo (China)	Social Media	14	Not available										
Reddit (United States)	Social News	13	No restrictions	No restrictions	Prohibited		Restricted—country laws and age requirements apply	Restricted—specific policy to protect under 18s	Restricted—country laws apply		No restrictions	No restrictions	Not applicable
Twitter (United States)	Social Media	13	No restrictions	No restrictions	Prohibited		Restricted—country laws and age requirements apply	Prohibited—specific policy to protect minors	Restricted—country laws and age requirements apply	Prohibited—specific policy to protect minors	No restrictions	No restrictions	Not applicable
Douban (China)	Social Media	Not available	Not available										
Snapchat (United States)	Social Media	13	Specific nutritional claim requirements apply		Prohibited		Restricted—country laws and age requirements apply	Restricted—specific policy to protect under 18s	Restricted		No restrictions	No restrictions	Not applicable
LinkedIn (United States)	Social Media	16	No restrictions	No restrictions	Prohibited		Restricted—country laws apply		Prohibited		Prohibited		Not applicable
Pinterest (United States)	Social Media	13	No restrictions	No restrictions	Prohibited		Restricted—country laws and age requirements apply	Restricted—specific policy to protect under 18s	Restricted—country laws apply		Restricted		Not applicable

^1^ Ordered by the number of active users as of October 2019 [26]; ^2^ Defined as people aged below 18 years old, unless specified otherwise.

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
