# Peer review of "The Advertising Policies of Major Social Media Platforms Overlook the Imperative to Restrict the Exposure of Children and Adolescents to the Promotion of Unhealthy Foods and Beverages"

_ijerph, 2020, doi:10.3390/ijerph17114172_

Round 1

Reviewer 1 Report

I would congratulate the author for the study. The subject of study is very relevant for the improvement of health of the population.

The article is written in an appropriate way, and data and analyses are appropriately presented, and results provide an advance in current knowledge of how major global social media platforms manage the advertising of unhealthy foods. However, no supplementary information of platforms’ advertising policies is needed because the information is summarized in Table 1. But the fact that the information is in a supplementary file makes its inclusion acceptable.

Publishing the study may question these platforms to adopt voluntary restrictions relating to the exposure of children to marketing of unhealthy foods and, which will benefit child and adolescents’ health and improve population diets.

Reviewer 2 Report

Thank you for the opportunity to review this interesting manuscript.

The information in the manuscript is likely to be of interest to readers of the journal, representing relevant advertising policies of social media platforms at a point in time (i.e. Dec 2019). Many factors leading to unhealthy lifestyles change over time, including advertising policies and exposure to marketing. The authors have made a case for action by the social media platform companies.

The activity is a desktop review and was unable to locate advertising policies for 25% of the 16 most popular platforms. The authors have not pursued this further (for example by contacting the companies and asking for a copy of the relevant policy). Being unable to find a policy does not mean that there is no policy, although it may imply that if a policy exists it is not publicly promoted. This missing data is a minor weakness of the manuscript which might be mentioned in the discussion.

There is no information on the exposure of people/children using the social media platform to unhealthy food and drink. This is a relevant concern for the authors since an implication is that children using a platform with a strong policy will have less exposure to advertising of the harmful food or substance. Not only does this appear to be unknown, but there is no information on how large a problem this is (other than by implication). Perhaps the authors could include analysis of food advertising on social media platforms as a direction for further research.

Reviewer 3 Report

The topic raised in the work is very important from the point of view of public health. Especially in light of the growing problem of childhood obesity and type 2 diabetes, or other nutritional disorders. The work is written correctly in terms of content, presentation of results and I have read it with interest.
However, I lacked reference how we should influence social media platforms to significantly limit the promotion of unhealthy food.
In addition, I believe that the title of the work does not fully reflect what the content describes. The article does not discuss the role of social media platforms in restricting the exposure of children and adolescent to the promotion of unhealthy foods and beverages - the title should rather sound like that: The necessity of implementing restrictions to the exposure of children and adolescent to the promotion of unhealthy foods and beverages through Social Media Platforms

Reviewer 4 Report

line 24 - Keywords: should not be identical to the subject

line 92-120 - in my opinion this section is disproportionately short compared to the introductory part. It should be expanded. I also suggest separating specific subsections.

line 141 - I'm not sure if any citation should be in the resulting part

line 147-152 - this part would better fit into the discussion

line 168-197 - in my opinion this section is disproportionately short - It should be expanded.

what were the limitations of the study?

Reviewer 5 Report

1st Review

Title:  The Potential Role of Social Media Platforms in 3 Restricting the Exposure of Children and Adolescents 4 to the Promotion of Unhealthy Foods and Beverages

Manuscript ID: ijerph-812554

The manuscript “The Potential Role of Social Media Platforms in  Restricting the Exposure of Children and Adolescents to the Promotion of Unhealthy Foods and Beverages” aimed to examine the advertising policies of major global social media platforms related to the advertising of unhealthy foods, and identify opportunities for social media platforms to take action to limit the exposure of children to marketing of unhealthy foods. The theme is interesting; however I have minor recommendations to improve the quality of the paper.

  • Some parts of the introduction section should be placed in the discussion section. You should highlight the hypothesis at the end of the introduction section.
  • The discussion section should be improved. In the case of this manuscript, join the results and discussion section could be welcome.
  • Since the social medias evaluated are in most countries around the world, it is important to discuss the different realities, as well as show the populations’ access to the internet/social media in countries.
  • In the conclusion section, you should highlight your main findings.

Round 2

Reviewer 4 Report

Thank you for the corrections. Good job :)